# The Impact of Age on Statin-Related Glycemia: A Propensity Score-Matched Cohort Study in Korea

**DOI:** 10.3390/healthcare10050777

**Published:** 2022-04-22

**Authors:** Shaopeng Xu, Seung-Woon Rha, Byoung Geol Choi, Hong Seog Seo

**Affiliations:** 1Cardiovascular Department, Tianjin Medical University General Hospital, Tianjin 300052, China; 2Cardiovascular Center, Division of Cardiology, Korea University Guro Hospital, Seoul 08308, Korea; mdhsseo@korea.ac.kr; 3Cardiovascular Research Institute, Korea University College of Medicine, Seoul 02841, Korea; trv940@naver.com

**Keywords:** statin, dysglycemia, age, atherosclerotic cardiovascular disease

## Abstract

The aim of this study was to investigate the influence of statin on glycemic control in different age groups. Patients admitted for suspected or confirmed coronary artery disease between January 2005 and December 2013 in Seoul, Korea were initially enrolled. After propensity score matching, 2654 patients (1:1 statin users and non-users) were selected out of total 5041 patients, including 1477 “young” patients (≤60 y) and 1177 elderly patients (>60 y). HbA1c was decreased by 0.04% (±0.86%) in statin non-users. On the contrary, a slight increment of 0.05% (±0.71%) was found in statin users (*p* < 0.001). The change patterns of HbA1c were constant in both young and elderly patient groups. Furthermore, elderly statin users demonstrated significantly worse glycemic control in serum insulin and homeostatic model assessment—insulin resistance (HOMA-IR) index. In elderly patients, statin users were found to have a 2.61 ± 8.34 μU/mL increment in serum insulin, whereas it was 2.35 ± 6.72 μU/mL for non-users (*p* = 0.012). Statin users had a 0.78 ± 3.28 increment in HOMA-IR, in contrast to the 0.67 ± 2.51 increment in statin non-users (*p* = 0.008). In conclusion, statin treatment was associated with adverse glycemic control in the elderly population.

## 1. Introduction

Statins are widely used for patients who are at risk of or have cardiovascular diseases, to lower their serum levels of low-density lipoprotein cholesterol (LDL), which is the presumed origin of atherosclerotic cardiovascular disease (ASCVD). Since the disclosure of Justification for the Use of Statins in Prevention: an Intervention Trial Evaluating Rosuvastatin (JUPITER) in 2008, showing that statin could increase the risk of new-onset diabetes (NOD), clinicians raised concern about the statin-associated adverse glucose control [1]. Recently published data demonstrated the dysglycemic effect of statins and statin therapy, especially high dose therapy, that resulted in increased diabetes risk, yet stain usage is increasing [1,2,3,4,5]. Indeed, it is well established that statin induces excess risk of type 2 diabetes [4]. This risk is particularly elevated in patients with already high diabetes risk, according to warning from the US Food and Drug Administration in 2012 [5]. However, the predictors of the development of diabetes have not been fully clarified. Given the evident benefit and the necessity of statin therapy, clarifying the vulnerable patients is critical to providing more care to them to reduce the possible development of diabetes and consequent complications. 

Being postmenopausal, having a high body mass index (BMI) and having a high baseline fasting blood glucose level are considered as risk factors of NOD under statin therapy [2,5,6,7]. However, to date, there is still no consensus on whether age is a risk factor of statin-associated dysglycemia [8,9,10,11,12,13]. Statin-related dysglycemia, e.g., new onset diabetes, may occur at younger ages, as shown by an IDEAL trial [2], and older ages [9,12,13]. Furthermore, it is not established whether the dosage of statin should be modified in elderly patients based on the possible incremental risk of dysglycemia. However, statin has been recommended for patients <75 years old at high risk of cardiovascular diseases in order to reduce the likelihood of ASCVD related events [4,14]. Given the inconsistent results of prior studies, the current study aimed to investigate the influence of statin on glycemic control.

## 2. Materials and Methods

### 2.1. Study Participants and Study Design

In this retrospective cohort study, a total of 5041 patients who were admitted for suspected or confirmed coronary artery disease between January 2005 and December 2013 at Guro Hospital, Seoul, Korea, were initially enrolled. The diagnosis criteria of suspected or confirmed coronary artery disease included: (1) had >50% coronary artery stenosis detected during coronary angiogram testing; (2) had typical symptoms of cardiac ischemia or (3) had myocardial infarction; (4) had coronary revascularization surgery, either Percutaneous Transluminal Coronary Intervention (PCI) or Coronary Artery Bypass Grafting (CABG). All patients signed patient informed consent and were divided into statin user and non-user groups. The statins strategy was left to the clinician’s discretion according to patient’s individual conditions. Patients were excluded if they changed their statin strategy plans, i.e., either statin users suspended statin use, or non-statin users started to use statins during the follow-up period, regardless of any reasons. 

### 2.2. Data Assessment

The following demographic and clinical data were extracted from medical records: age at prescription date, gender, diagnostic history of acute coronary syndrome, diabetes and impaired glucose tolerance; concurrent medications, including β-blockers, calcium-channel blockers (CCB), angiotensin II receptor blockers (ARB), angiotensin-converting enzyme inhibitors (ACEI), diuretics, nitrates and fenofibrate. Laboratory testing results before prescription of statin were also extracted, including serum levels of fasting blood glucose, glycosylated hemoglobin (HbA1c), total cholesterol, high-density lipoprotein (HDL), triglyceride, LDL, high-sensitivity C-reactive protein (hs-CRP), aspartate aminotransferase (AST), alanine aminotransferase (ALT), alkaline phosphatase (ALP), fasting insulin, creatinine and uric acid. The homeostasis model assessment—insulin resistance (HOMA-IR) index was also calculated, which was defined as fasting glucose (mmol/L) × fasting insulin (μU/mL) divided by 22.5 as a parameter of insulin resistance [15].

Patients were divided into two groups according to their statin administration (statin user and non-user). Besides, patients were further classified into a “young” group (≤60 years old) and an elderly group (>60 years old). 

### 2.3. Statistical Analysis

Continuous variables at baseline were compared between two groups using an independent-sample Student’s *t*-test and are presented as means ± standard deviations (std). Categorical variables at baseline were compared between two groups using chi-square statistics and are each presented as a number with a percentage. Between-group differences of the changes in fasting blood glucose, fasting insulin, HbA1c and HOMA-IR after statin treatment were compared using a repeated general linear model (GLM), while adjusting for gender, BMI, diabetes, β-blocker use, ACEI use, ARB use and diuretics use. 

In order to balance the baseline characteristics between statin users and non-users, propensity score matching was utilized. Those baseline characteristics that were significantly different between statin user and non-user groups, including age, sex, hyperlipemia, diabetes, impaired glucose tolerance and coronary artery disease, were adjusted for propensity score, with a caliper width of 0.02, and neighbor 1:1 matching algorithm. A *p*-value < 0.05 (two sides) was considered as statistically significant. Data were analyzed using SPSS software (version 19.0 Inc., Chicago, IL, USA).

## 3. Results

### 3.1. Cohort Baseline and Glycemic Control before Propensity Score Matching 

Initially, 5041 patients were enrolled, the majority of whom (*n* = 3541, 70.2%) were treated with a statin. Only 1500 (29.8%) patients were not receiving any statin therapy (Table 1, Appendix A). Compared with non-users, statin users were more likely to be male (51.4%) and older (60.55 ± 10.42 years), and had higher prevalences of hyperlipidemia (14.1%), diabetes (55.4%) and coronary artery disease (11.8%) (all *p*-value < 0.05). On the other hand, less stain users had impaired glucose tolerance than non-statin users (32.8% vs. 43.5%). Furthermore, statin users also had higher serum levels of fasting blood glucose (120.02 ± 38.41 vs. 111.13 ± 34.64 mg/dL in statin users and non-users, respectively), HbA1c and insulin; a higher mean HOMA-IR; and were more likely to receive ACEI or ARB (all *p*-value < 0.05). Nearly all these differences were consistent in the subgroup analysis stratified by age. 

In the overall cohort, after treatment, statin users (−0.03 ± 0.98%) had a 50% lower level of HbA1c (−0.02 ± 0.82%), a 21.2% lower insulin level and a 13.4% lower level of HOMA-IR than non-statin users (all *p* < 0.05) (Table 2). These significant variances were also observed in the subgroup of elderly patients. However, in young patients, statin users (−0.02 ± 1.06) only had a more significant reduction in serum level of HbA1c than young statin non-users (−0.00 ± 0.79) (*p* < 0.001). 

### 3.2. Cohort Baseline and Glycemic Control after Propensity Score Matching 

After propensity score matching, statin users and non-users were not statistically different in gender, age, the prevalence of dyslipidemia, diabetes, coronary artery disease, glucose metabolism profile (including fasting blood glucose, insulin level, HbA1c and HOMA-IR) and concomitant medications (all *p* > 0.05) (Table 3). However, statin users had a higher serum level of LDL than non-users (108.08 ± 38.42 and 113.10 ± 27.02 mg/dL for statin users and non-users, respectively), but lower Lpa (26.69 ± 29.33 and 19.80 ± 19.44 mg/dL) (all *p* < 0.001). 

In the subgroups of young and elderly patients, lower Lpa levels were also observed in statin users, regardless of age group. No statistical differences were observed in young patients in LDL level between statin users and non-users (*p* = 0.559). Nevertheless, in elderly patients, statin users performed better in lipid control for total cholesterol and LDL (all *p* < 0.001). 

Fasting blood glucose was increased by 0.54 ± 30.98 mg/dL in statin non-users and 2.24 ± 30.27 mg/dL in statin users, although this discrepancy was not statistically significant (*p* = 0.292) (Table 4, Figure 1). More importantly, HbA1c was decreased by 0.04% (±0.86%) in statin non-users, whereas an increment in 0.05% (±0.71%) was found in statin users (*p* < 0.001) (Figure 2). The change patterns of HbA1c were constant in both young and elderly patient groups. Serum levels of fasting insulin and HOMA-IR were all increased among statin users and non-users in the total population and the different age groups (Figure 3). However, in the elderly patient group only, statin users showed worse performance in these indicators. In this subgroup, statin users were found to have a 2.61 ± 8.34 μU/mL increment in serum insulin, whereas the non-users had a 2.35 ± 6.72 μU/mL (*p* = 0.012) increment. There was a 0.78 ± 3.28 increment in HOMA-IR among users, and a 0.67 ± 2.51 increment in non-users (*p* = 0.008) (Table 4, Figure 4).

## 4. Discussion

In the current cohort, statin treatment increased HbA1c significantly in all patients, regardless of age group. In addition, statin use was associated with adverse glucose metabolism, signified by increments in insulin level and HOMA-IR in the elderly patients but not in young patients. 

Previous results have suggested an excess risk of new onset diabetes of around 9–13% in individuals treated with statins [9,14,15,16]. This off-target effect could be triggered through several mechanisms: inhibition of insulin secretion from pancreatic beta-cells by means of changes in activity of voltage-gated calcium channels; reduction in serum ubiquinol-10 and consequently of ATP production and insulin secretion; insulin resistance of peripheral tissues due to attenuation of glucose transporter 4 expression on adipocytes and damage to skeletal muscle cells [15,17,18,19,20,21].

A study conducted by Sattar et al. supported the increased risk of NOD in the elderly statin uses [9]. In this meta-analysis, which identified 13 statin trials with 91,140 participants, Sattar et al. reported a higher risk of developing diabetes in elderly statin users than younger statin users. The result was consistent with outcomes reported in a recent study conducted by Ma et al. and the ATTEMPT trial [12,13]. Conversely, after pooled TNT analysis in IDEAL and SPARCL studies, Waters and his colleagues did not find age differences between patients with and without NOD [2]. A study conducted by Chen et al. drew a similar conclusion to the IDEAL study: that the risk of statin-related NOD was more evident for young women (aged 40–64 years old) compared to the elderly (aged 65 years or more) [10]. The adjusted risk increased about 14% for every 10-year increment in age, taking 40–54 as a baseline age range. Increased age exhibited an inverse relationship with the risk of statin-induced NOD. Meanwhile, the third kind of inconsistent result was reported by several previous studies [3,11], which suggested no predictive effect of age or only a non-significant numeric trend of younger age predisposing to statin-related NOD.

Different types of statin were believed to exert different dysglycemic effect in patients with or without diabetes. Through comparison studies on effects of several statins, Cortese F et al. concluded that rosuvastatin should represent the first-choice drug in the management of diabetic subjects, not only due to its lipid lowering efficacy in these patients, but also due to its pleiotropic effects [22]. A 20 mg rosuvastatin daily dose for 12 weeks is effective at reducing discomforts in subjects with diabetic polyneuropathy [23]. Additionally, more importantly, no significant change in fasting glucose, glycated hemoglobin or nerve growth factor beta was found. As to the risk of NOD development during statin treatment, it was related to the type of statin and the intensity of dose administered [22]. Pravastatin, one of the less powerful statins but one which has only minor diabetogenic effects, should be preferred in the management of patients with high LDL-C serum levels who are at low CV risk and have predictive factors for diabetes. Conversely, rosuvastatin, known as the most powerful statin for reducing serum LDL-C but which has a significant effect on the development of NOD, should be used in subjects at high CV risk or in secondary prevention. In regard to the dose to administer, physicians should take into account the patient’s past medical history. Besides, they should perform periodic long-term glycometabolic control in order to eventually switch to a lower statin dose.

Notably, the demographic characteristics and concomitant medication were not equal between NOD cases and controls in the study reported by Chen et al. [10]. Higher prevalences of ACEI and triglyceride-lowering medications were observed in control groups. Recently, several studies reported the impact of antihypertensive medicine on glucose metabolism, which has been accepted widely and could not be ignored. Furthermore, the baseline BMI was not available, either, which was deemed as an independent risk factor of NOD. The above-mentioned studies [12,13], which found a higher incidence of NOD in the elderly participants, could not demonstrate the impact of age on the statin-related dysglycemic effect, even though they proved the predictive effect of age on NOD. Moreover, the uncertainty of comparable baseline characteristics between groups of different ages attenuated the strength of the analyses [10,12,13]. 

Considering the improvements via ACEI or ARB and the deterioration caused by diuretics or β-blockers on glycemic metabolism/control, which have been confirmed by clinical investigations [24,25,26], the combination medication was matched with propensity score and adjusted with GLM in the present study. In addition to antihypertensive medicine, in the present study, we matched the fenofibrate use between the statin user and non-user groups with propensity score as well. The existing evidence suggested that combination statin/fibrate therapy has a potential neutralizing effect on the adverse pro-diabetic impact of statins and could prevent NOD [27]. After matching and adjusting the confounding factors, the current analysis demonstrated a statistically significant increment in HbA1c and numerical increments in fasting blood glucose, insulin level and HOMA-IR in the statin user group compared with the non-user group, for the whole cohort, which confirmed the statin-related dysglycemia. Furthermore, in different age subgroups, the significant increments in the elderly group compared with the insignificance of the increments in young patient group in terms of glucose metabolism parameters suggested the deteriorated glucose control induced by statin was mainly driven by older age. This evidence might imply that age could worsens the statin-related dysglycemic effect. 

The significantly higher level in HbA1c of 0.09% induced by statin treatment in the current study was in line with previous studies, confirming the statin-related dysglycemic effect [20,28,29,30]. In the meta-analysis conducted by Erqou et al., a higher mean HbA1c level by 0.12% was observed in participants treated with statin than in controls [30]. Our findings of non-significant increments in fasting blood glucose, insulin and HOMA-IR after statin use in the present analysis are consistent with the results of some previous studies [31,32,33,34]. One of the possible reason for the lack of impact of statin on these parameters is the mixture of diabetes and non-diabetes patients. In patients with diabetes, the value of HOMA-IR and insulin level for evaluating insulin sensitivity is still controversial [34]. In nondiabetic patients, statin use was associated with significant increases in insulin level, fasting blood glucose and HOMA-IR [35], whereas no significant increase was observed in diabetes [36]. Therefore, in the present analysis, which involved a 36.3% diabetic cohort, the non-significant change induced by statin could be reasonable. The second reason could have been the insensitivity of fasting glucose as an index of glucose metabolism compared to postprandial blood glucose. Despite statin’s detrimental effect on glucose control and the increment in risk of NOD, fasting blood glucose was similar between statin-treated patients and controls, as in the JUPITER study [1]. 

Due to the increased risk of dysglycemia in the elderly compared to youth, in regard to effective lipid control, modifications of statin dosage and being more alert to NOD should be considered. Based on these observations in the present analysis, we suggest more monitoring of insulin resistance in nondiabetic patients who are receiving statin therapy, and monitoring blood glucose concentrations in elderly patients with diabetes who are receiving statin therapy.

Several limitations shall be noted. One limitation is that the present analysis is a retrospective study. Potential confounders were therefore inevitable, and selection bias cannot be eliminated. The baseline was not equal between statin users and non-users. However, the propensity score matching was performed to minimize the confounding factors. After matching, the baseline characteristics were similar for statin users and non-users. Furthermore, we did not evaluate the dysglycemic effect of statin among nondiabetic patients in particular. At baseline, a large portion of patients had impaired glucose tolerance. Further study should be encouraged to evaluate the dysglycemic effect of statins in populations free from glucose-related impairments. Lastly, we did not analyze the impacts of statin dosage and individual statin type on the glycemic control, which has been explored in previous study [37], showing the different potentials according to different types and dosages of statin regarding increasing the risk of NOD. A further large prospective trial to verify the age effect on statin-related dysglycemia is warranted.

## 5. Conclusions

In brief, statin treatment was associated with adverse glycemic control, which was characterized by increased HbA1c. Dysglycemia was more evident in elderly statin users than elderly non-users. Older age could have strengthened the detrimental effect of statin on glycemic control.

## Figures and Tables

**Figure 1 healthcare-10-00777-f001:**
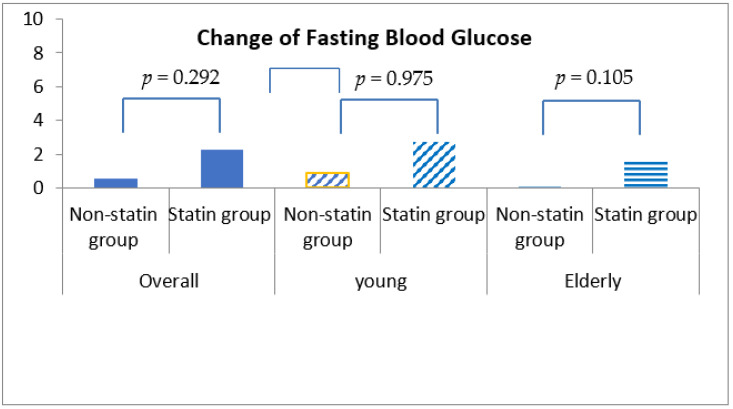
Changes in fasting blood glucose levels among non-statin and statin users overall and in young and elderly cohorts after propensity score matching.

**Figure 2 healthcare-10-00777-f002:**
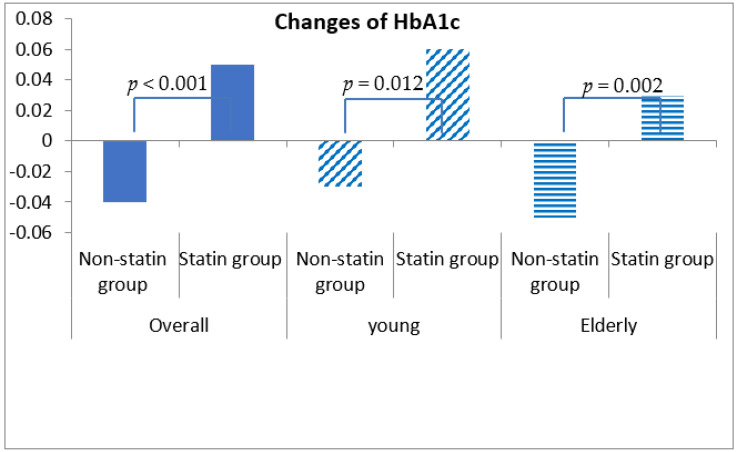
Changes in serum levels of HbA1c among non-statin and statin users overall and in young and elderly patients after propensity score matching.

**Figure 3 healthcare-10-00777-f003:**
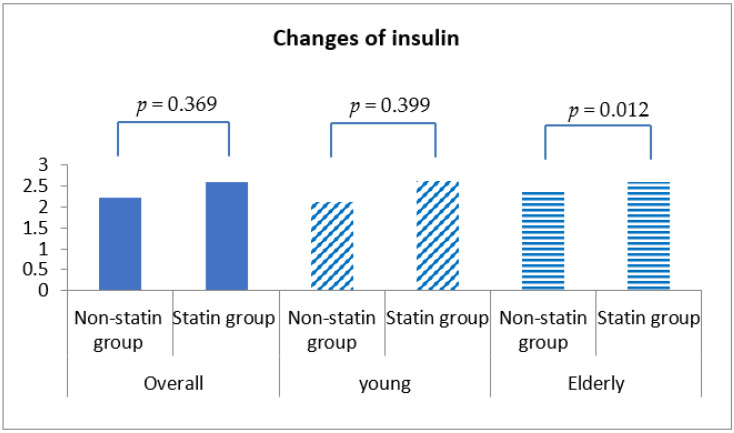
Changes in serum insulin level among non-statin and statin users overall and in young and elderly patients after propensity score matching.

**Figure 4 healthcare-10-00777-f004:**
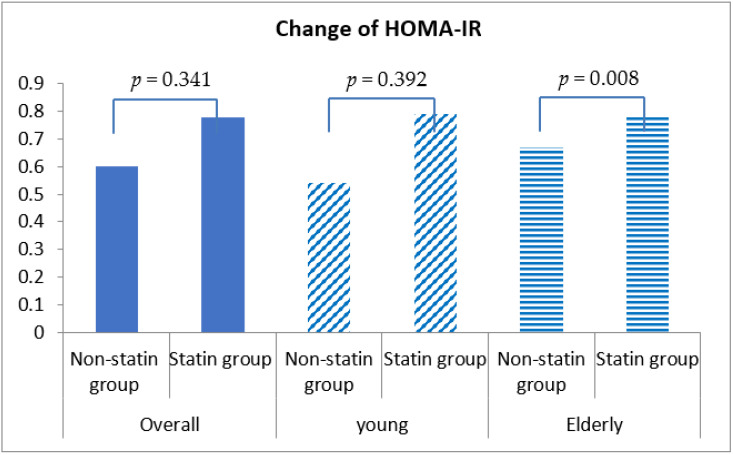
Changes in homeostasis model assessment insulin resistance (HOMA-IR) index among non-statin and statin users overall and in young and elderly patients after propensity score matching.

**Table 1 healthcare-10-00777-t001:** Baselines of study patients stratified by age group before propensity score matching.

Variables	Overall (*n* = 5041)			Young, ≤60 y (*n* = 2571)			Elderly, >60 y (*n* = 2470)		
	Non-Statin Group (*n* = 1500)	Statin Group (*n* = 3541)	*p*-Value	Non-Statin Group (*n* = 890)	Statin Group (*n* = 1681)	*p*-Value	Non-Statin Group (*n* = 610)	Statin Group (*n* = 1860)	*p*-Value
**Baseline characteristics**									
Gender (male), *n* (%)	715 (47.7%)	1821 (51.4%)	0.015	453 (50.9%)	1010 (60.1%)	<0.001	262 (43.0%)	811 (43.6%)	0.815
Age (year), mean ± std	56.97 ± 11.49	60.55 ± 10.42	<0.001	49.46 ± 7.80	51.81 ± 6.74	<0.001	67.93 ± 5.78	68.45 ± 5.82	0.051
Hyperlipidemia, *n* (%)	155 (10.3)	499 (14.1)	<0.001	94 (10.6%)	265 (15.8%)	<0.001	61 (10.0%)	234 (12.6%)	0.098
Diabetes, *n* (%)	472 (31.5%)	1962 (55.4%)	<0.001	255 (28.7%)	972 (57.8%)	<0.001	393 (64.4%)	870 (46.8%)	<0.001
Impaired glucose tolerance, *n* (%)	652 (43.5%)	1160 (32.8%)	<0.001	384 (43.1%)	485 (28.9%)	<0.001	268 (43.9%)	675 (36.3%)	0.001
Cardiovascular disease, *n* (%)	6 (0.4%)	418 (11.8%)	<0.001	2 (0.2%)	168 (10.0%)	<0.001	4 (0.7%)	250 (13.4%)	<0.001
**Laboratory findings**									
BMI ^†^ (kg/m^2^), mean ± std	24.57 ± 3.34	24.69 ± 2.94	0.416	24.55 ± 3.37	24.85 ± 2.81	0.125	24.61 ± 3.30	24.56 ± 3.04	0.820
Total cholesterol (mg/dL), mean ± std	178.74 ± 30.51	169.73 ± 43.42	<0.001	179.71 ± 30.09	174.71 ± 44.14	0.001	177.32 ± 31.09	165.23 ± 42.27	<0.001
HDL ^†^ -choleterol (mg/dL), mean ± std	51.98 ± 13.8	50.64 ± 13.13	0.001	51.90 ± 13.42	50.56 ± 12.78	0.013	52.10 ± 14.43	50.71 ± 13.44	0.030
LDL ^†^ -cholesterol (mg/dL), mean ± std	113.03 ± 26.77	104.35 ± 39.14	<0.001	113.18 ± 26.15	107.95 ± 39.89	<0.001	112.79 ± 27.68	101.10 ± 38.17	<0.001
Triglyceride (mg/dL), mean ± std	145.34 ± 109.47	145.81 ± 97.88	0.885	152.11 ± 124.35	155.82 ± 117.85	0.457	135.46 ± 82.22	136.77 ± 74.30	0.713
hs-CRP ^†^ (mg/L), mean ± std	2.57 ± 7.07	2.92 ± 9.21	0.194	2.12 ± 5.17	2.58 ± 7.12	0.087	3.24 ± 9.12	3.22 ± 10.75	0.978
Fasting glucose (mg/dL),	111.13 ± 34.64	120.02 ± 38.41	<0.001	111.06 ± 36.78	123.57 ± 39.94	<0.001	111.25 ± 31.30	116.81 ± 36.69	0.001
HbA1c ^†^ (%), mean ± std	6.15 ± 1.14	6.65 ± 1.26	<0.001	6.06 ± 1.11	6.68 ± 1.34	<0.001	6.27 ± 1.16	6.63 ± 1.19	<0.001
Insulin (μU/mL), mean ± std	8.23 ± 6.35	9.41 ± 7.05	<0.001	8.33 ± 6.32	9.53 ± 7.19	<0.001	8.07 ± 6.39	9.30 ± 6.92	<0.001
HOMA-IR ^†^	2.36 ± 2.37	2.89 ± 2.74	<0.001	2.41 ± 2.49	3.03 ± 2.89	<0.001	2.29 ± 2.19	2.77 ± 2.61	<0.001
Lpa (mg/dL), mean ± std	19.91 ± 19.46	26.58 ± 28.11	<0.001	19.14 ± 18.98	24.97 ± 26.80	<0.001	20.99 ± 20.11	27.73 ± 28.96	<0.001
AST ^†^ (IU/L), mean ± std	25.61 ± 18.96	26.45 ± 17.47	0.137	25.82 ± 21.78	27.34 ± 19.40	0.075	25.30 ± 13.85	25.64 ± 15.50	0.640
ALT ^†^ (IU/L), mean ± std	24.71 ± 21.86	26.25 ± 18.29	0.019	26.95 ± 25.88	29.69 ± 21.34	0.005	21.68 ± 13.43	23.14 ± 14.32	0.030
ALP ^†^ (IU/L), mean ± std	70.65 ± 21.37	71.33 ± 24.34	0.571	68.85 ± 20.03	69.88 ± 24.54	0.491	74.01 ± 23.35	72.99 ± 24.02	0.608
Creatinine (mg/dL), mean ± std	0.81 ± 0.21	0.83 ± 0.34	0.001	0.78 ± 0.18	0.80 ± 0.25	0.029	0.85 ± 0.24	0.87 ± 0.42	0.137
Uricacid(mg/dL), mean ± std	5.18 ± 1.49	5.10 ± 1.50	0.118	5.15 ± 1.50	5.11 ± 1.46	0.545	5.23 ± 1.47	5.10 ± 1.54	0.087
**Concurrent medications**									
Β blocker, *n* (%)	382 (25.5%)	895 (25.3%)	0.887	210 (23.6%)	327 (19.5%)	0.014	172 (28.2%)	568 (30.5%)	0.285
CCB ^†^, *n* (%)	533 (35.5%)	1251 (35.3%)	0.897	274 (30.8%)	433 (25.8%)	0.007	259 (42.5%)	818 (44.0%)	0.541
ARB ^†^, *n* (%)	533 (35.5%)	1371 (38.7%)	0.033	291 (32.7%)	563 (33.5%)	0.692	242 (39.7%)	808 (43.4%)	0.109
ACEI ^†^, *n* (%)	146 (9.7%)	545 (15.4%)	<0.001	79 (8.9%)	260 (15.5%)	<0.001	67 (11.0%)	285 (15.3%)	0.008
Diuretics, *n* (%)	533 (35.5%)	1023 (28.9%)	<0.001	264 (29.7%)	360 (21.4%)	<0.001	269 (44.1%)	663 (35.6%)	<0.001
Fenofibrate, *n* (%)	47 (3.1%)	79 (2.2%)	0.075	26 (2.9%)	57 (3.4%)	0.559	21 (3.4%)	22 (1.2%)	0.001

^†^ BMI, body mass index, HDL, high-density lipoprotein; LDL, low-density lipoprotein; hs-CRP, high-sensitivity c-reactive protein; HbA1c: glycosylated hemoglobin; HOMA-IR, homeostatic model assessment; AST, glutamic oxalacetic transaminase; ALT, glutamic-pyruvic transaminase; ALP, alkaline phosphatase; CCB: calcium-channel blockers; ARB: angiotensin II receptor blockers; ACEI: angiotensin-converting enzyme inhibitors.

**Table 2 healthcare-10-00777-t002:** Changes in serum levels of glucose, HbA1c, insulin and HOMA-IR among non-statin users and statin users before propensity score matching—overall and in young and elderly cohorts.

Variables	Overall (*n* = 5041)			Young, ≤60 y (*n* = 2571)			Elderly, >60 y (*n* = 2470)		
	Non-Statin Group (*n* = 1500)	Statin Group (n = 3541)	*p*-Value ^†^	Non-Statin Group (*n* = 890)	Statin Group (n = 1681)	*p*-Value ^†^	Non-Statin Group (*n* = 610)	Statin Group (*n* = 1860)	*p*-Value ^†^
Fasting blood glucose, (mg/dL), mean ± std	0.69 ± 29.31	0.63 ± 38.21	0.650	1.07 ± 29.90	1.22 ± 38.49	0.992	0.14 ± 28.45	0.10 ± 37.97	0.769
HbA1c ^‡^, (%), mean ± std	−0.02 ± 0.82	−0.03 ± 0.98	<0.001	−0.00 ± 0.79	−0.02 ± 1.06	<0.001	−0.05 ± 0.86	−0.04 ± 0.90	<0.001
Insulin (μU/mL), mean ± std	2.50 ± 6.79	1.97 ± 8.75	0.005	2.57 ± 6.83	1.71 ± 8.15	0.860	2.39 ± 6.72	2.21 ± 9.26	<0.001
HOMA-IR ^‡^	0.67 ± 2.46	0.58 ± 4.09	0.006	0.65 ± 2.44	0.51 ± 4.10	0.861	0.68 ± 2.50	0.64 ± 4.07	<0.001

^†^ Analyzed with the repeated general linear model (GLM) on the basis of adjustment for gender, body mass index (BMI), diabetes, angiotensin-converting enzyme inhibitors (ACEI), angiotensin II receptor blockers (ARB), diuretics and β-blocker. ^‡^ HbA1c: glycosylated hemoglobin; HOMA-IR, homeostatic model assessment.

**Table 3 healthcare-10-00777-t003:** Baselines of study patients stratified by young and elderly groups after propensity score matching.

Variables	Overall (*n* = 2654)			Young (*n* = 1477)			Elderly (*n* = 1177)		
	Non-Statin Group (*n* = 1327)	Statin Group (*n* = 1327)	*p*-Value ^†^	Non-Statin Group (*n* = 723)	Statin Group (*n* = 754)	*p*-Value ^†^	Non-Statin Group (*n* = 604)	Statin Group (*n* = 573)	*p*-Value ^†^
**Baseline Clinical Characteristics**									
Gender (male), *n* (%)	635 (47.9%)	624 (47.0%)	0.697	373 (51.6%)	419 (55.6%)	0.130	242 (40.1%)	232 (40.5%)	0.506
Age (year), mean ± std	58.51 ± 10.87	58.40 ± 10.46	0.798	50.61 ± 7.17	51.25 ± 7.00	0.081	67.97 ± 5.79	67.82 ± 5.74	0.650
Hyperlipidemia, *n* (%)	149 (11.2%)	149 (11.2%)	1.000	88 (12.2%)	96 (12.7%)	0.753	61 (10.1%)	52 (9.1%)	0.555
Diabetes, *n* (%)	469 (35.3%)	495 (37.3%)	0.313	252 (34.9%)	293 (38.9%)	0.118	217 (35.9%)	202 (35.3%)	0.855
Impaired glucose tolerance, *n* (%)	569 (42.9%)	596 (44.9%)	0.309	306 (42.3%)	303 (40.2%)	0.429	263 (43.5%)	293 (51.1%)	0.110
Cardiovascular disease, *n* (%)	6 (0.5%)	6 (0.5%)	1.000	2 (0.3%)	3 (0.4%)	1.000	4 (0.7%)	3 (0.5%)	1.000
**Laboratory findings**									
BMI ^†^ (kg/m^2^), mean ± std	24.62 ± 3.33	24.90 ± 3.08	0.115	24.64 ± 3.35	24.97 ± 2.99	0.160	24.58 ± 3.32	24.79 ± 3.20	0.433
Total cholesterol (mg/dL), mean ± std	179.00 ± 30.95	174.35 ± 41.93	0.001	180.40 ± 30.70	179.45 ± 42.82	0.623	177.31 ± 31.19	167.64 ± 39.79	<0.001
HDL ^†^ -choleterol (mg/dL), mean ± std	51.68 ± 13.99	52.20 ± 13.07	0.323	51.35 ± 13.56	52.05 ± 12.52	0.308	52.07 ± 14.47	52.40 ± 13.78	0.691
LDL ^†^ -cholesterol (mg/dL), mean ± std	113.10 ± 27.02	108.08 ± 38.42	<0.001	113.40 ± 26.39	112.38 ± 39.05	0.559	112.74 ± 27.77	102.40 ± 36.85	<0.001
Triglyceride (mg/dL), mean ± std	147.98 ± 112.96	144.60 ± 92.36	0.399	158.13 ± 132.37	152.31 ± 105.17	0.350	135.82 ± 82.52	134.45 ± 70.95	0.760
hs-CRP ^†^ (mg/L), mean ± std	2.67 ± 7.30	2.21 ± 5.55	0.067	2.21 ± 5.25	2.20 ± 5.42	0.990	3.22 ± 9.14	3.11 ± 5.71	0.823
Fasting blood glucose (mg/dL), mean ± std	112.74 ± 35.96	113.25 ± 33.17	0.704	113.92 ± 39.32	115.30 ± 36.25	0.492	111.33 ± 31.44	110.56 ± 28.43	0.656
HbA1c ^†^ (%), mean ± std	6.23 ± 1.17	6.29 ± 1.08	0.173	6.20 ± 1.17	6.28 ± 1.14	0.197	6.27 ± 1.17	6.31 ± 1.00	0.528
Insulin (μU/mL), mean ± std	8.42 ± 6.47	8.61 ± 5.89	0.413	8.69 ± 6.51	8.62 ± 5.81	0.803	8.09 ± 6.41	8.62 ± 6.00	0.145
HOMA-IR ^†^	2.45 ± 2.46	2.49 ± 2.00	0.686	2.58 ± 2.66	2.55 ± 2.08	0.785	2.29 ± 2.20	2.40 ± 1.89	0.357
Lpa (mg/dL), mean ± std	19.80 ± 19.44	26.69 ± 29.33	<0.001	18.65 ± 18.66	25.08 ± 27.67	<0.001	21.06 ± 20.20	28.53 ± 31.06	<0.001
AST ^†^ (IU/L), mean ± std	26.03 ± 19.72	25.68 ± 15.07	0.614	26.62 ± 23.49	26.13 ± 18.11	0.661	25.32 ± 13.90	25.08 ± 9.75	0.740
ALT ^†^ (IU/L), mean ± std	25.29 ± 22.92	25.76 ± 18.92	0.571	28.29 ± 28.16	28.18 ± 21.67	0.937	21.70 ± 13.46	22.60 ± 13.97	0.270
ALP ^†^ (IU/L), mean ± std	71.24 ± 21.02	70.89 ± 29.34	0.829	69.72 ± 19.37	69.98 ± 30.79	0.902	73.68 ± 23.27	72.56 ± 26.45	0.669
Creatinine (mg/dL), mean ± std	0.81 ± 0.22	0.82 ± 0.30	0.545	0.78 ± 0.19	0.80 ± 0.30	0.085	0.85 ± 0.24	0.83 ± 0.31	0.455
Uricacid (mg/dL), mean ± std	5.18 ± 1.47	5.07 ± 1.46	0.073	5.13 ± 1.46	5.10 ± 1.47	0.679	5.23 ± 1.47	5.04 ± 1.46	0.228
**Concurrent medications**									
Β blocker, *n* (%)	324 (24.4%)	325 (24.5%)	1.000	152 (21.0%)	143 (19.0%)	0.329	172 (28.5%)	182 (31.8%)	0.227
CCB ^†^, *n* (%)	467 (35.2%)	498 (37.5%)	0.226	211 (29.2%)	225 (29.8%)	0.819	256 (42.4%)	273 (47.6%)	0.079
ARB ^†^, *n* (%)	474 (35.7%)	501 (37.8%)	0.295	234 (32.4%)	250 (33.2%)	0.782	240 (39.7%)	251 (43.8%)	0.174
ACEI ^†^, *n* (%)	137 (10.3%)	140 (10.6%)	0.899	70 (9.7%)	82 (10.9%)	0.493	67 (11.1%)	58 (10.1%)	0.636
Diuretics, *n* (%)	441 (33.2%)	475 (35.8%)	0.178	178 (24.6%)	210 (27.9%)	0.174	263 (43.5%)	265 (46.2%)	0.379
Fenofibrate, *n* (%)	42 (3.2%)	52 (3.9%)	0.345	29 (4.0%)	36 (4.7%)	0.534	21 (3.5%)	13 (2.3%)	0.228

^†^ BMI, body mass index, HDL, high-density lipoprotein; LDL, low-density lipoprotein; hs-CRP, high-sensitivity c-reactive protein; HbA1c: glycosylated hemoglobin; HOMA-IR, homeostatic model assessment; AST, glutamic oxalacetic transaminase; ALT, glutamic-pyruvic transaminase; ALP, alkaline Phosphatase; CCB: calcium-channel blockers; ARB: angiotensin II receptor blockers; ACEI: angiotensin-converting enzyme inhibitors.

**Table 4 healthcare-10-00777-t004:** Changes in serum levels of fasting blood glucose, HbA1c, insulin and HOMA-IR index among non-statin and statin users after propensity score matching, overall and in young and elderly cohorts.

Variables	Overall (*n* = 2654)			Young, ≤60 y (*n* = 1477)			Elderly, >60 y (*n* = 1177)		
	Non-Statin Group (*n* = 1327)	Statin Group (*n* = 1327)	*p*-Value ^†^	Non-Statin Group (*n* = 723)	Statin Group (*n* = 754)	*p*-Value ^†^	Non-Statin Group(*n* = 610)	Statin Group (*n* = 573)	*p*-Value ^†^
Fasting blood glucose, (mg/dL), mean ± std	0.54 ± 30.98	2.24 ± 30.27	0.292	0.91 ± 32.88	2.75 ± 31.33	0.975	0.09 ± 28.57	1.56 ± 28.82	0.105
HbA1c ^‡^, (%), mean ± std	−0.04 ± 0.86	0.05 ± 0.71	<0.001	−0.03 ± 0.86	0.06 ± 0.73	0.012	−0.05 ± 0.87	0.03 ± 0.68	0.002
Insulin (μU/mL), mean ± std	2.23 ± 6.62	2.61 ± 7.37	0.369	2.13 ± 6.53	2.62 ± 6.54	0.399	2.35 ± 6.72	2.61 ± 8.34	0.012
HOMA-IR ^‡^	0.60 ± 2.50	0.78 ± 2.78	0.341	0.54 ± 2.50	0.79 ± 2.34	0.392	0.67 ± 2.51	0.78 ± 3.28	0.008

^†^ Analyzed with repeated general linear model (GLM) on the basis of adjustments for gender, body mass index (BMI), diabetes, angiotensin-converting enzyme inhibitors (ACEI), angiotensin II receptor blockers (ARB), diuretics and β-blocker. ^‡^ HbA1c: glycosylated hemoglobin; HOMA-IR, homeostatic model assessment.

## Data Availability

The datasets generated and/or analyzed during the current study are not publicly available due to ethical restrictions, but are available from the corresponding author on reasonable request.

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
