# Peer review of "The Impact of Age on Statin-Related Glycemia: A Propensity Score-Matched Cohort Study in Korea"

_healthcare, 2022, doi:10.3390/healthcare10050777_

Round 1
Reviewer 1 Report
This paper investigated the impact of statin use on glycemic control in younger and older age groups and found that statin use was associated with worse glycemic control, predominantly in older people. The paper was very well-written and easy to follow. The data were also well and clearly presented. I commend the authors' work and only have some minor suggestions for authors' consideration. Please feel free to take or leave it.
1. In introduction section, the authors mentioned that it is still under debate about whether age is a risk factor of statin-associated dysglycemia. It would also be important to add what’s the impact of hyperglycemia on older people’s health in particular.
2. In the introduction, the authors wrote “statin has been recommended for elderly patients with or without cardiovascular diseases to reduce the risks of ASCVD related events”. This is not correct. Among people aged younger than 75 years, statins are recommended to use in people with high CVD risk; Among those aged over 75 years, the net benefits of statins remains uncertain and no relevant recommendation in guidelines is made.
3. The difference in baseline characteristics between statin and non-statin users is significant before propensity-score matching. Did the author conduct goodness-of-fit test for the propensity score model?
4. Could the author add a couple of sentences somewhere in the Discussion Section to explain why statins may increase glucose/HbA1c level, and why there might be a difference between younger and older people (e.g. difference in glucose metabolism, older people are more susceptible to statin related side effect?)?
5. In the limitation, the authors mentioned they didn’t evaluate the dysglycemic effect of statin in nondiabetic patients in particular. Could the authors explain why? Taken it from me it is important to do a subgroup analysis by the presence of diabetes, given the heterogeneity in glucose metabolism/control between diabetic and non-diabetic people.
Author Response
Please find our detailed responses in the attached Reply Letter. Many thanks.

Reviewer 2 Report
Title: “The impact of age on statin-related glycemia: a propensity score-matched cohort study in Korea”
I read this manuscript and I think that:
- The English of the paper should be improved. Please consider the help of a native English speaker.
- The retrospective nature of this study can be considered as a limitation. This should be discussed in a dedicated limitation section.
- At baseline, 31.5% of patients suffer with diabetes, 43.5% with impaired glucose tolerance. How did these conditions impact in results? I think that a population free from glucose metabolism impairment would have been more reliable.
- All comorbidities should be described and include in the final analysis. Please update.
- Please discuss the paper from Cortese F et al. Pharmacol Res. 2016 May;107:1-18.
Author Response

(The authors gave the same response as above.)

Reviewer 3 Report
I congratulate the authors on taking up the interesting topic of the impact of statins on glycemia control in different age groups. The study group is large. It includes 5041 patients. The study itself lasted 8 years. The topic is interesting and timeless.
I propose to use several issues:
1. On what specific basis were patients qualified to two groups, i.e. to the group of suspected and confirmed patients with coronary artery disease.
I propose to supplement this in Materials and Methods.
2. I propose to make a flowchart for the distribution of study groups to be presented as Fig. 1.
3. I propose to present in Figure 2 baseline of study patients stratified by young and elderly group after propensity score matching
4. However, I propose to add a subsection of study limitations
as a retrospective study
Author Response

(The authors gave the same response as above.)

Round 2
Reviewer 1 Report
I appreciate the authors for addressing all my comments.
Reviewer 3 Report
Thank you, no comments